# DMEM and EMEM as alternate growth media for pathogenic *Leptospira*

Leandro E. Garcia[1,2], Zitong Lin[2], Sophie Culos[3], M. Catherine Muenker[2], Emily E. Johnson[2], Zheng Wang[4], Francesc Lopez-Giraldez[5], Alexandre Giraud-Gatineau[6], Angela M. Jackson[7], Mathieu Picardeau[6], David R. Goodlett[3,7], Jeffrey P. Townsend[4,8], Helena Pĕtrošová[3,7], Elsio A. Wunder Jr[1,2,9]*

1 Department of Pathobiology and Veterinary Science, University of Connecticut, Storrs, Connecticut, United States of America, 2 Department of Epidemiology of Microbial Diseases, Yale School of Public Health, New Haven, Connecticut, United States of America, 3 Department of Biochemistry and Microbiology, University of Victoria, Victoria, British Columbia, Canada, 4 Department of Biostatistics, Yale School of Public Health, New Haven, Connecticut, United States of America, 5 Yale Center for Genomic Analysis, New Haven, Connecticut, United States of America, 6 Institut Pasteur, Université Paris Cité, CNRS UMR, Biology of Spirochetes Unit, Paris, France, 7 University of Victoria Genome BC Proteomic Center, Victoria, British Columbia, Canada, 8 Department of Ecology and Evolutionary Biology, Yale University, New Haven, Connecticut, United States of America, 9 Gonçalo Moniz Institute, Oswaldo Cruz Foundation, Brazilian Ministry of Health, Salvador, Brazil

* elsio.wunder@uconn.edu

## Abstract

Pathogenic *Leptospira* species can survive and thrive in a wide range of environments. Distinct environments expose the bacteria to different temperatures, osmolarities, and amounts and sources of nutrition. However, leptospires are mostly cultured in a laboratory setting under *in vitro* conditions that do not reflect natural environments. This constraint on laboratory cultures limits the applicability of *in vitro* studies to the understanding of even simple pathogenic processes. Here we report, investigate, and identify a medium and conditions that mimic the host environment during leptospirosis infection, expanding the available *in vitro* tools to evaluate leptospiral pathogenesis. We quantified genome-wide transcription of pathogenic *Leptospira interrogans* cultured in different *in vitro* media compositions and conditions—EMJH at 29 °C and DMEM, EMEM, and HAN at 37 °C and 5% $CO_2$. Using EMJH as standard, we compared gene expression in these compositions to genome-wide transcription gathered in a host environment: whole blood (WB) of hamsters after infection with pathogenic leptospires. Leptospires cultured in DMEM and EMEM media shared 40% and 47% of all differentially expressed genes (DEGs) of leptospires present within WB (FDR<0.01), while leptospires cultured in HAN media only shared 20% of DEGs with those from WB. Furthermore, gene and pathway expression of leptospires cultured on DMEM and EMEM media exhibited a better correlation with leptospires grown in WB, including promoting expression of a similar leptospiral lipid A profile to the one identified directly in host tissues. Taken together, these results indicate that commercial cell-culture media EMEM or DMEM are better surrogates for *in vivo* pathogenic studies than EMJH or HAN media in *Leptospira*. These alternative culture

**Data availability statement:** RNA-Seq reads are available in the NCBI GEO data base (GSE297028). Mass spectrometry imaging data were converted to the imzML format in SCiLS (centroid, no normalization). The resulting imzML files can be accessed via METASPACE using the following link: https://metaspace2020.eu/project/Leptospira_lipidA_liver.

**Funding:** This work was supported by NIH grants R21AI163663 (EAWJ), R01AI182354 (EAWJ), and R01AI147314 (DRG), a capacity grant from The Ambrose Monell Foundation (EAWJ), and a discovery grant RGPIN-2022-04433 by Natural Sciences and Engineering Research Council of Canada (DRG). The work performed at the University of Victoria-Genome BC Proteomics Centre was also supported by funding to the Metabolomics Innovation Centre (TMIC) from Genome Canada and Genome British Columbia, through the Genomics Technology Platform (GTP) program for operations and technology development (265MET, DRG), as well operations support from the Canadian Foundation for Innovation Major Sciences Initiative (CFI-MSI) program (35456, DRG). Infrastructure and operations funding to support this project was provided by PacifiCan project 22591 (DRG). The funders had no role in study design, data collection and analysis, decision to publish, or preparation of the manuscript.

**Competing interests:** I have read the journal's policy and the authors of this manuscript have the following competing interests: DRG is a co-founder and a vice president of Patagain, a company that develops mass spectrometry-based microbiology tests to identify pathogens and determine antimicrobial resistance. The company was not involved in the development of this work or the analysis and interpretation of the results. The other authors acknowledge no competing interests while working and preparing this manuscript.

conditions, using media that are a standard supply worldwide, provide a reproducible and cost-effective approach that can accelerate research investigation and reduce the number of animal infections necessary for basic research of leptospirosis.

## Author's summary

Leptospirosis is a life-threatening disease that can occur in diverse epidemiological settings and is the leading zoonotic disease in terms of morbidity and mortality worldwide. However, the major burden and impact of this neglected disease is in the world's poorest countries. The burden of leptospirosis is likely to increase, as extreme weather events intensify in frequency with global climate change. The complexity of the genus and the tools to culture and isolate leptospires, especially on developing countries, thwarted research and the development of sensitive diagnostic assays and effective prevention methods. We evaluated different media and culture conditions to grow leptospires *in vitro*, including spirochetes harvest directly from the blood of infected hamsters, and compared to the traditional *in vitro* culturing of leptospires in EMJH medium at 30 °C. Using transcriptomic analysis and lipid A profiling, our results indicated that the growth of leptospires at 37 °C with 5% $CO_2$ in EMEM and DMEM, traditional mammalian cell culture media, was not only feasible but also induced an overall gene expression and a lipid A profile similar to the one directly observed in the host environment. EMEM and DMEM have a standard composition, without expensive and complex components, and are available worldwide at affordable cost. These alternative media can provide researchers with reliable culture conditions for leptospires, facilitating research and its reproducibility, while accelerating discoveries in the field for this neglected disease worldwide.

## Introduction

The use of laboratory animals in research is a common and often highly informative practice. A diverse field of studies have been using research animals to better understand pathogenesis, evaluate vaccine candidates, or viability of new drugs and treatments. However, research animals often experience disagreeable symptoms and painful disease progression [1,2]. Animal experiments can also have a distressing effect on personnel performing this work [3]. These drawbacks have spurred increased animal welfare and bioethical regimentation [4]. Indeed, the importance of animal use in research has been debated; legislation has been enforced to protect animals and maintain humane conditions while conducting such research. Efforts have been encouraged to explore alternatives to animal use, including the 3Rs: replacement, reduction, and refinement [3,4]. One possible alternative to reduce animal use in research is the expansion of *in vitro* assays that mimic specific conditions, like temperature and nutrition, that are highly relevant factors in the host environment [5,6], enabling the generation *in vivo* of essential information regarding pathogenesis without infecting animals.

Animal models such as guinea pigs, hamsters, mice, and rats are widely used for pathogenesis studies of leptospirosis [7], a zoonotic disease of global importance. Leptospirosis is caused by a diverse group of pathogenic spirochetes of the genus *Leptospira*, an imposes a significant burden on impoverished populations living in tropical climates [8]. It is estimated that the disease causes 1 million cases with a mortality rate of ~60,000 deaths per year worldwide [8]. A large spectrum of reservoir animals, from rats to wild animals, can survive infection and have their renal tubules colonized with leptospires. Live pathogenic *Leptospira* can then be excreted in urine, contaminating environments (soil, mud, and water) where the spirochete can survive for weeks, causing spillover infections to humans and other animals and contributing to the life cycle of the disease [9–11]. There are no effective diagnostics or vaccines for leptospirosis, hampering prevention and treatment efforts and leading to infections with lethality as high as 10–50% [12]. The burden of leptospirosis is believed to increase with the rise of the urban slum population worldwide [13,14] and extreme weather events intensified by global climate change [15]. The lack of understanding of the molecular mechanisms of leptospirosis pathogenesis has curtailed the development of effective control measures for this important neglected disease [16].

The most common medium used and commercially available worldwide for culture of leptospires, EMJH, was developed by Ellinghausen and McCullough [17] and modified by Johnson and Harris [18]. The EMJH medium replaced previously used serum-containing media (e.g., Stuart and Korthof), which fostered leptospiral growth but caused phosphate precipitates complicating darkfield microscope visualization [19]. EMJH typically consists mostly of a salt base supplemented with bovine serum albumin (fraction V), Tween 80, chlorides, sulfides, vitamins, and rabbit serum, with some minor variations in formulation [20–22]. EMJH balanced and crafted composition based on nutritional and metabolic needs of the bacteria, makes it an ideal medium for the growth of several species of pathogenic and nonpathogenic *Leptospira*. Puzzlingly, *in vitro* growth of leptospires commonly occurs is in aerobic conditions at 29–30 °C—a temperature that is not consistent with host environments and physiologies. A medium that has a composition similar to the host and can be used to culture *Leptospira* in host-like conditions can induce patterns of protein expression that are similar to those observed during the natural infection process [23–25]. The balanced salt solution of media like Minimum Essential Medium Eagle (MEM), when added to EMJH, changed the osmolarity and affected leptospiral regulation [26]. Recently, Hornsby-Alt-Nally developed the HAN medium, a modified EMJH formulation for propagation of leptospires; HAN facilitated increased growth of leptospires directly from clinical samples [27]. The main features of the HAN medium are the use of hemin as an iron source and the addition of Ham's Nutrient mixture F12 (Dulbecco's Modified Eagle's Medium), which enable the culture and growth of leptospires at 37 °C and 5% $CO_2$.

The specific nutrition formula and growth conditions of a medium can be crucial to the capability of individual leptospiral strains surviving or adapting to culture. However, little is known about the distinct transcriptomics and biological pathways that are altered by those different media formulations. To understand the adaptation of leptospires to culture conditions and nutrition, we analyzed their transcriptomic profiles. Using RNA-seq, we compared the transcriptome of leptospires during the process of infection and in response to cell culture media that are commonly used for the growth of leptospires or mammalian cells. We then applied mass spectrometry approaches to compare *Leptospira* lipid A profiles under these different culturing conditions and in host tissue. Comparing these profiles to those associated with leptospires in a whole-blood hamster model, we identified a culture medium and conditions that can be used as a surrogate for *in vivo* studies on leptospires. Use of these media for studies of leptospiral pathogenesis increases the applicability of *in vitro* virulence studies, while also saving costs and research time and reducing the need for animal infections to elucidate basic properties of leptospiral pathogenesis.

## Results

### Media composition and culturing conditions determine growth behavior in pathogenic leptospires without affecting their virulence

We evaluated the growth of pathogenic *Leptospira* strain Fiocruz L1-130 in different media. To establish the optimal timepoint for comparing transcriptomes during the mid- to late-log phase of bacterial growth, we characterized the growth

curve. We observed environmental effects using a single strain: distinct behaviors in each medium (Fig 1A). Growth in DMEM and EMEM media (37 °C) trended similarly: leptospires exhibited rapid exponential growth that peaked on day 4, followed by a longer stationary phase through day 10, with a decline on the number of cells after that. Culture in EMJH (29 °C) and HAN (37 °C) led to a long exponential phase with peaks at days 10 and 14, respectively. However, leptospires cultured on HAN media never reached the concentrations observed in EMJH (29 °C), plateauing at similar concentrations as DMEM and EMEM. Finally, transcriptome analysis of *Leptospira* genome-wide gene expression on the EMJH medium at 37 °C was not possible: these culture conditions were not conducive to the growth and multiplication of leptospires (Fig 1A), as observed previously [27]. Previous studies that successfully used leptospires cells cultured in EMJH at 37 °C either performed an overnight shift from 30 °C or used higher inoculum doses compared to the low dose of inoculum ($10^4$ leptospires) used in our experiments [28–30]. Furthermore, we evaluated the virulence of our *Leptospira* strain and its ability to cause disease in the hamster model after *in vitro* culture using each medium. All animals were challenged using a dose of $10^8$ leptospires by a conjunctival route and reached the endpoint criteria between 8- and 10-days post-challenge (Fig 1B). Similar to previous results obtained with *L. interrogans* cultured using HAN [31,32], culture in these diverse media, temperatures, and conditions did not negatively affect the growth and virulence of the strain upon infection.

## Whole blood as a surrogate for host environment in leptospirosis infection

Pathogenic leptospires can multiply and disseminate in the blood of hamsters. Dissemination that leads to colonization of significant organs and clinical symptoms [33]. Accordingly, we selected blood as a surrogate for the host environment relevant for infection. We then determined the best component of the hamster blood (whole blood, plasma, or serum) to be used to assess the transcriptome of *Leptospira* during infection. With an inoculum dose of $10^8$ leptospires of strain Fiocruz L1-130 by conjunctival route, the animals started presenting symptoms of severe leptospirosis between days 8–10

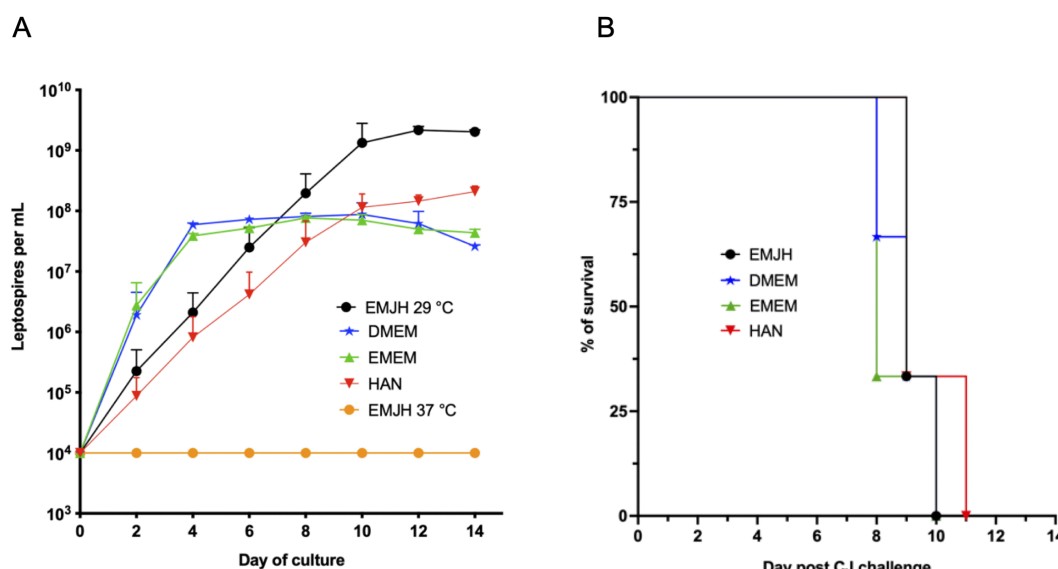

**Fig 1. Growth and virulence analysis of pathogenic *Leptospira* cultured on different media and conditions. A.** Growth curve of *L. interrogans* serovar Copenhageni strain Fiocruz L1-130 in five different growing conditions (medium, temperature, and $CO_2$ level). Dots and error bars represent the average from two independent experiments and standard deviation, respectively. **B.** Survival curve of virulence experiments. *Leptospira* strain Fiocruz L1-130 was cultured in DMEM, EMEM, or HAN at 37 °C and 5% $CO_2$, or EMJH at 30 °C. Groups of hamsters (n = 3) were challenged with $10^8$ leptospires by conjunctival-route (CJ) to demonstrate the ability of pathogenic *Leptospira* to maintain its virulence after culture on different media and conditions. Results presented are representative of one of two independent experiments with similar results.

post-challenge [34], at which point the animals were euthanized. At this stage, leptospires in the blood are on their peak and antibodies start to circulate [35]. Therefore, for this study we collected blood from the animals on day seven after the challenge to represent leptospires in the host environment.

Our qPCR analysis revealed statistically significant differences in the bacterial burden among the blood components. Leptospire concentration in whole blood reached $2.0 \times 10^6$ leptospires genome equivalents (GEq)/mL, which was significantly higher ($P < 0.05$) than the concentration reached in serum and plasma ($7.9 \times 10^3$ and $2.4 \times 10^3$ leptospires GEq/mL, respectively; S1 Fig). Therefore, we selected whole blood seven days post-challenge as the comparator sample to evaluate media that approximate *Leptospira* in their host environment.

### The transcriptome profile of pathogenic *Leptospira* is modulated by different culture conditions, with growth at 37 °C and 5% CO$_2$ eliciting gene transcription closely related to the host environment

We analyzed the complete transcriptomic data from different treatments using EMJH conditions as a baseline. The time for RNA extraction of leptospires on culture was defined based on the late-logarithmic stage under each media condition: day 4 for DMEM and EMEM and day 10 for HAN and EMJH at 29°C (Fig 1A). Principal component analysis (PCA) showed a strong association between our biological duplicates for each condition. Furthermore, a clear separation between EMJH and WB (62% of the variance in PC1) was observed, with EMEM and DMEM samples clustered closer to WB and HAN samples clustered closer to EMJH (Fig 2A). Differentially expressed genes (DEGs) were identified by a threshold False Discovery Rate (FDR) of less than 0.01 and an absolute log2 Fold Change (log2FC) ≥ 2 (S1 Table). These thresholds established a total of 414 DEGs in DMEM, 505 DEGs in EMEM, 271 DEGs in HAN, and 740 DEGs in whole blood tissue (WB) of hamsters infected with strain L1-130 (Fig 2B), indicating that temperature and the composition of the medium exhibited significant effects on the transcriptome of pathogenic leptospires. Hierarchical clustering and heatmap analysis, using selected DEGs associated with virulence mechanisms [36–46], exhibited similarly distinct patterns of expression, with a clear division of the EMJH cluster compared to other media and WB samples (Fig 2C). Also, in comparison to EMJH, we observed a similar gene transcription pattern across WB with EMEM, DMEM, and HAN media (surrogate media). Genes regulating important virulence factors or regulatory networks (e.g., *hem0*, *ligA*, *ligB*, *sph2*, *colA*, *Tolc*, and *perB*) [42,47–50] were upregulated in WB and surrogate media when compared to EMJH. In contrast, other virulence-related genes (e.g., *lsa27*, *lsa25*, *loa22*, *lipl41*, and *lipl45*) [48,51], were significantly downregulated in WB and surrogate media compared to EMJH (Fig 2C). This observation indicates a high expression similarity between leptospiral gene transcription in WB and in the surrogate *in vitro* media (EMEM, DMEM, and HAN) when compared to EMJH. This pattern was further supported by analysis of the top 10 DEGSs (up and down regulated) in WB samples (S2 Table).

We then investigated the influence of *in vitro* surrogate media on the bacterial cells compared to the host environment, performing an enrichment analysis of biological pathways using BioCyc tools [52] on DEGs between all treatments. We identified similar biological pathways related to metabolism in the surrogate *in vitro* media and WB (S2-S7 Tables). All surrogate *in vitro* media evaluated induced downregulation of genes involved in fatty-acid and lipid degradation while upregulating genes related to the biosynthesis of cobalamin (vitamin B12). The DMEM medium induced higher expression levels for all analyzed genes in the vitamin B12 biosynthesis pathway, when compared with WB and other media (S2 Fig). Nevertheless, these results indicate that EMEM, DMEM, and HAN media, compared to EMJH, are good surrogates to study gene and pathway expressions that occurs during *in vivo* infection.

Quantitative real-time PCR of randomly selected genes (LIC10771, LIC11352, LIC11467, LIC11848, LIC20148, LIC_RS14805, and LIC_RS21185) in all conditions, corroborated our RNAseq results with a correlation coefficient of 0.70 (S3 Fig).

### DMEM and EMEM media are an *in vitro* alternative that mimics the host environment

To identify the similarity between our treatments regarding gene transcription, we performed a Pearson correlation of the log raw values of all DEGs (Fig 3A). We observed that WB samples exhibited the lowest correlation coefficients to EMJH

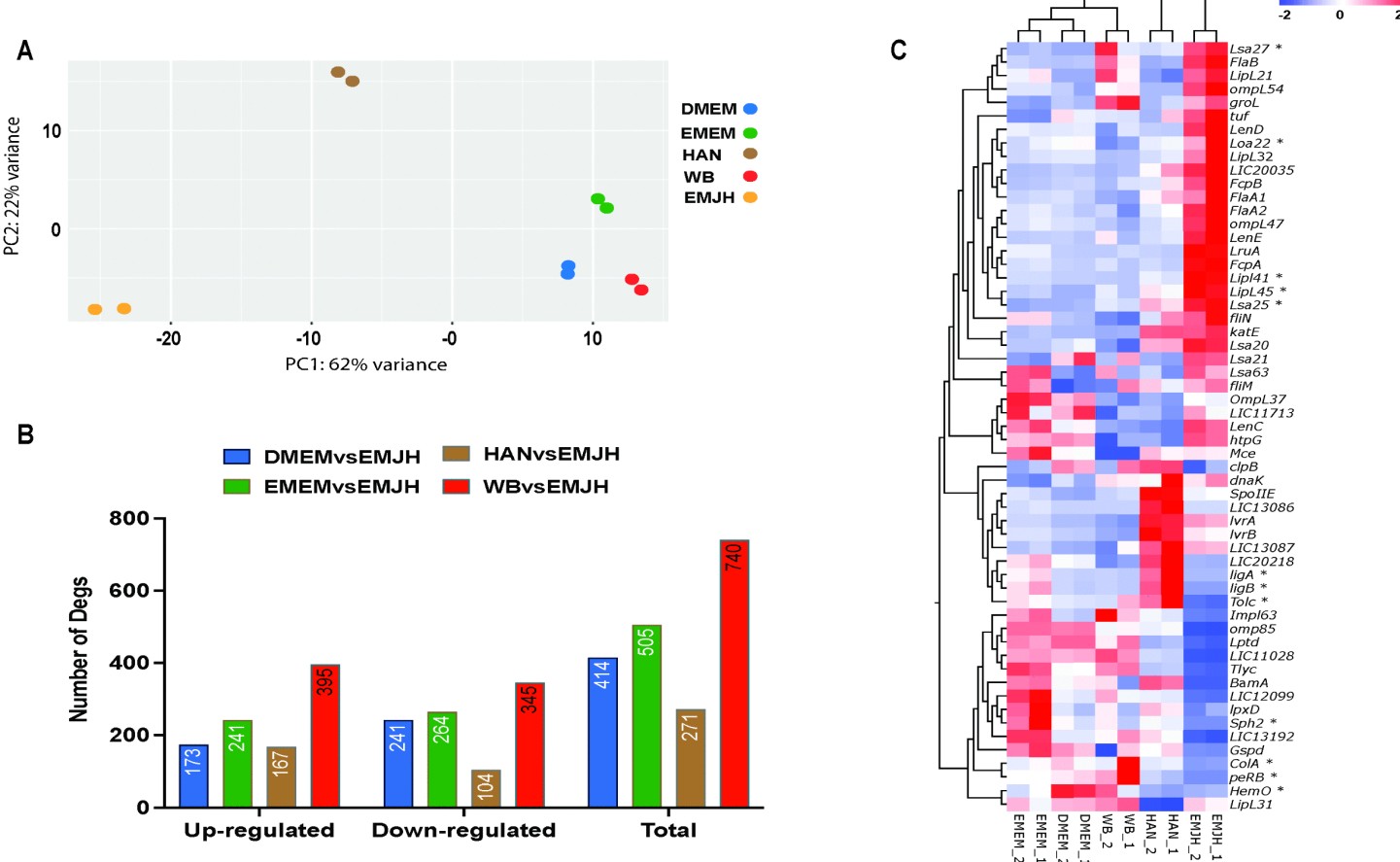

**Fig 2. Whole transcriptome and hierarchical cluster between different treatments. A.** First two Principal Component Analysis (PCA) between replicates of all the treatments evaluated: EMJH, EMEM, DMEM, HAN, and Whole-blood (WB). **B.** Total number of differentially expressed genes (DEGs) between all treatments versus EMJH. **C.** Hierarchical cluster of different treatments based on the expression of selected genes associated with the pathogenic mechanism of leptospirosis infection. * $P$ value < 0.01, and log2FC ± 2 (n = 2).

(0.51-0.56) and HAN (0.67-072). Cultured on EMEM exhibited a positive correlation coefficient with cultures on DMEM (0.96 and 0.97), while cultures on both media shared the highest positive correlation values with WB samples (DMEM: 0.84–0.86; EMEM: 0.84-0.87). We then performed a cluster of orthologous genes (COG) classification to categorize all differential expressed genes identified in WB (host environment) according to their predicted biological role (Fig 3B). Using EMJH as baseline, our analysis showed that EMEM and DMEM samples shared 164 and 132 DEGs with predicted biological function with the WB samples, respectively. In contrast, the HAN media induced only 64 COG-related DEGs shared with WB.

Furthermore, our DEGs analysis showed that leptospires cultured in EMEM and DMEM media shared more genes that were significantly up- (Fig 4A and 4B and S8 Table) and down-regulated (Fig 4C and 4D and S8 Table) in common with those in WB-cultured leptospires. Looking at the intersection of WB with DMEM and EMEM in the Venn diagram and UpSet plots, we noticed that DMEM and EMEM-cultured leptospires shared 79% and 72% of their up-regulates genes, respectively, with those up-regulated in WB-cultured leptospires (Fig 4A and 4B and S8 Table). Additionally, DMEM- and EMEM-cultured leptospires shared 65% of their leptospiral downregulated genes with those downregulated in

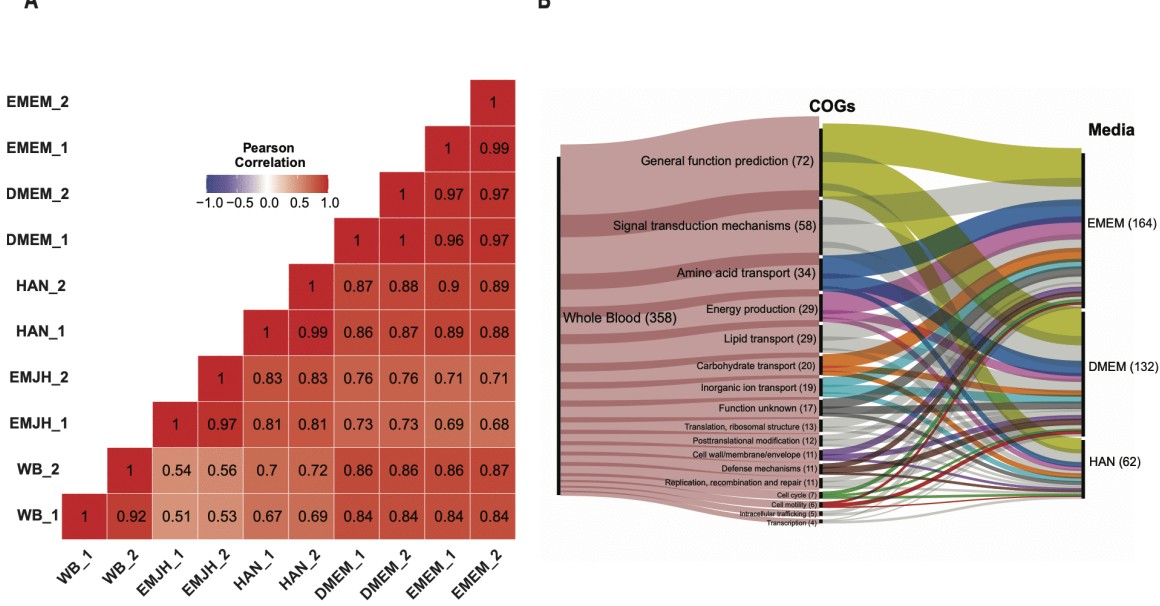

**Fig 3. Correlation and cluster of orthologous genes analysis. A.** Pearson correlation of normalized log raw counts of all differentially expressed genes (DEGs) between different growth conditions (*P* value <0.01). **B.** Sankey plot of orthologous gene clusters (COG) identified in *Leptospira* isolated from whole blood (WB) samples and shared with *Leptospira* grown in different *in vitro* media.

WB-leptospires (Fig 4C and 4D and S8 Table). Most importantly, when considering all DEGs identified in WB-leptospires, DMEM- and EMEM-leptospires shared 40% and 47% of them, respectively (S8 Table). In contrast, although cultured at 37°C with 5% $CO_2$, the HAN-leptospires shared only 18% of all the DEGs identified in WB-leptospires (Fig 4 and S8 Table). Those results indicate a higher similarity of DMEM- and EMEM-leptospires with the leptospiral gene transcription occurring during *in vivo* infection.

We identified 183 up- and 139 down-regulated genes in WB, compared to EMJH, that were not significantly regulated in any other media (S9 Table). This total represents 43.5% of all DEGs in WB. Most of those genes (51.5%) are hypothetical or poorly characterized proteins. Most likely those are genes involved in function and processes that are not being captured during *in vitro* conditions used here, including cell adhesion, translocation, immune evasion, colonization, etc. We saw up-regulation in the WB of 11 tRNA genes and transferases, involved in protein synthesis while some motility genes were downregulated. No major known pathogenic gene was present in that list, with exception of *loa22* which was significantly downregulated on WB, compared to EMJH. However, the trend of *loa22* expression is similar for the other surrogate media (Fig 2C), with similar trends for other targets as well. This result shows that the complexity of the host-pathogen interaction cannot be completely reproduced *in vitro*, but similar phenotypes to *in vivo* conditions can be achieved with a more suitable medium.

### *In vitro* culturing conditions at 37°C with 5% $CO_2$ induces a host-like expression of leptospiral lipid A biosynthesis genes

To further evaluate the impact of growth conditions on *Leptospira* cells, we investigated their outer membrane components. Lipopolysaccharide (LPS) is an important virulence factor in pathogenic *Leptospira*. It consists of three parts: O-antigen, core antigen, and a hydrophilic anchor to the *Leptospira* outer membrane, lipid A [53]. *L. interrogans* derived from the host tissue presents with an altered O-antigen when compared to *L. interrogans* grown under the traditional conditions

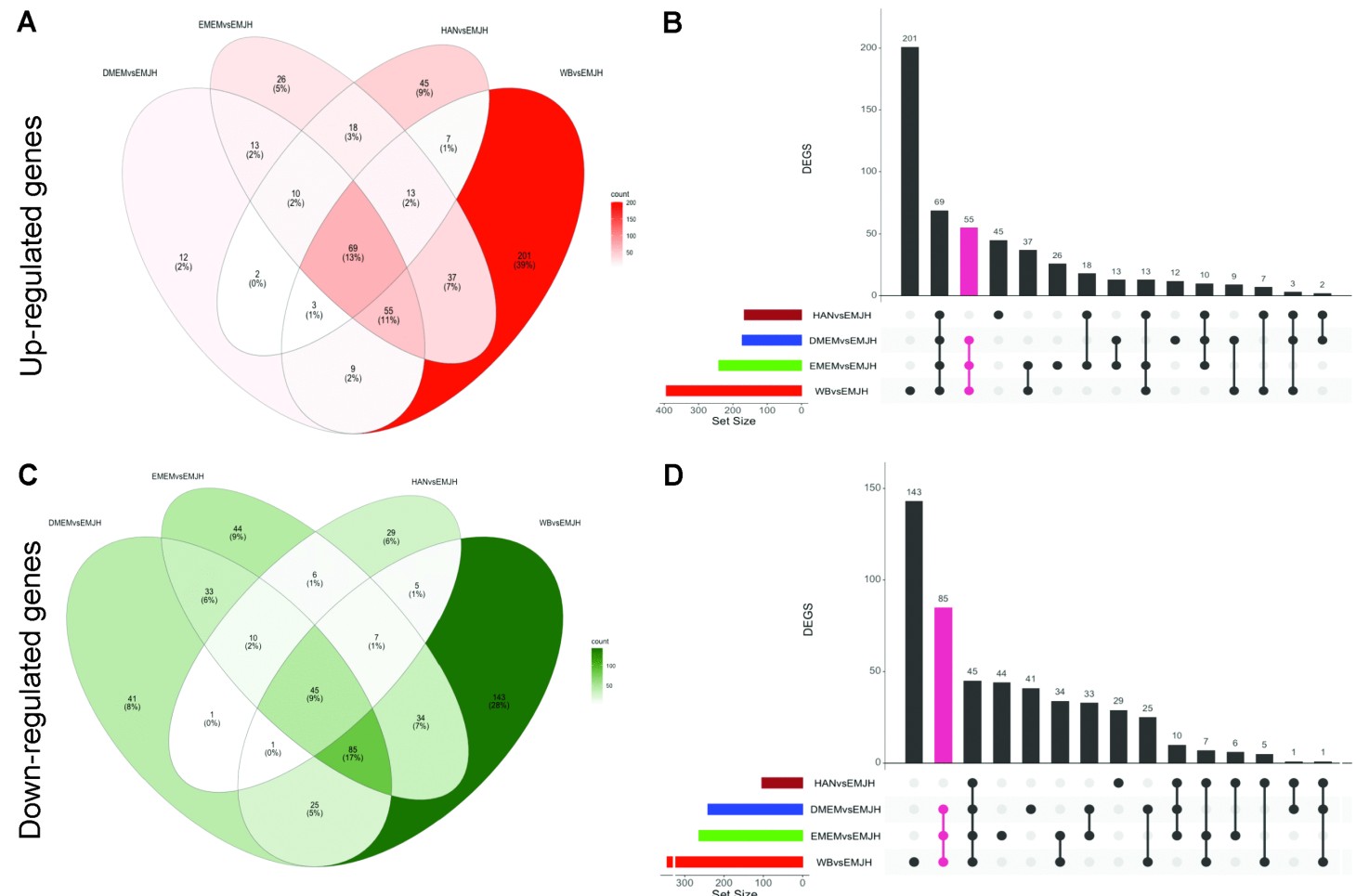

**Fig 4. Venn diagram and UpSet plot of differentially expressed genes (DEGs) between all treatments versus EMJH. A.** Venn diagram of up-regulated DEGs. **B.** Venn diagram of down-regulated DEGs. **C.** Upset plot of up-regulated DEGs. **D.** Upset plot of down-regulated DEGs. *P* value < 0.01, and log2FC ± 2 (n = 2).

*in vitro* (EMJH) [54]. The precise structure of the complex *L. interrogans* O-antigen has not been elucidated to date. Therefore, we focused on the effect of the growth conditions on *Leptospira* lipid A. Bacterial cells have the ability to modify their lipid A structure to facilitate environmental adaptation or to evade host immune response [55]. Lipid A biosynthesis is a multistep process facilitated by a family of *lpx* genes. To evaluate the potential influence of the *in vivo* and *in vitro* environment on lipid A synthesis, we compared the expression of 8 genes related to lipid A biosynthesis among our different grow conditions. The expression trend for most of those genes was similar between leptospires isolated from WB, HAN, EMEM, and EMEM compared to EMJH (S10 Table). The LpxC enzyme catalyzes a committed, energetically unfavorable, step-in lipid A biosynthesis [56]. The expression of *lpxC* was significantly upregulated (log2FC>2, *P*<0.05) in WB-leptospires and upregulated with low expression values in leptospires cultured in EMEM, DMEM, and HAN (S10 Table). LpxK phosphorylates the lipid A precursor to form lipid IV$_A$ in the last step of lipid A biosynthesis before adding sugar moieties to complete LPS [56]. Interesting, this gene was significantly upregulated in leptospires isolated from the host and leptospires cultured in DMEM, EMEM, and HAN. This result indicate that lipid A biosynthesis genes are altered in our *in vitro* media when compared to the EMJH medium, resulting in expression levels close related to the host environment.

### *L. interrogans* lipid A profile in DMEM and EMEM media is similar to the one directly observed in the host environment

Given the *in vitro* media-induced changes to expression of genes involved in biosynthesis of lipid A, we investigated the corresponding lipid A profiles. The lipid A of *L. interrogans* serovars Manilae and Copenhageni grown in EMEM, DMEM, HAN, and EMJH was extracted using the Fast Lipid Analysis Technique (FLAT) [57,58] and analyzed using MALDI mass spectrometry (Fig 5A–5E). Additionally, FLAT was used to investigate serum of *L. interrogans* serovar Copenhageni-infected hamsters. Lipid A profile of *L. interrogans* grown in EMJH at 30ºC corresponded to that published previously [43,59–61]. The lipid A profile was complex, with several lipid A ion clusters separated by 28 Da, and a base peak at *m/z* 1722 [59,60] (Fig 5A). Starting with a higher inoculum enabled us to obtain lipid A profile of *Leptospira* grown in the EMJH at 37 ºC with 5% $CO_2$. *L. interrogans* grown in EMJH and HAN at those conditions displayed an additional small lipid A ion cluster with a base peak at *m/z* 1542 (Fig 5B and 5C). Fragmentation *via* tandem MS analysis identified this cluster to consist of penta-acylated lipid A species. In comparison, lipid A profile of *L. interrogans* grown in EMEM and DMEM was significantly less complex (Fig 5D and 5E). The main lipid A cluster had a base peak at *m/z* 1748, corresponding to hexa-acylated lipid A species. The penta-acylated lipid A species with a base peak at *m/z* 1542 were also present (Fig 5D and 5E) under these conditions. *L. interrogans* serovars Copenhageni and Manilae share the same lipid A biosynthesis machinery, and we observed that their lipid A profiles were similar under different *in vitro* conditions (S4 Fig). No lipid A signal was detected in the serum of infected hamsters, likely due to low *Leptospira* abundance in this blood component (S4 Fig).

To reveal *Leptospira* lipid A profile directly from the host, liver of hamsters infected with *L. interrogans* serovar Manilae were investigated by MALDI mass spectrometry imaging. Given the similarity between the whole-genome and lipid A profile *in vitro* between serovars Copenhageni and Manilae, the result of the latter was determined as representative of *L. interrogans* species. Tissues were collected one-, two-, three-, and four-days post-challenge. Lipid A signal was detected on day three and reached maximum intensity on the experimental endpoint, at day four post-challenge (Fig 5F). The *Leptospira* lipid A profile in tissue (Fig 5G) resembled that identified in EMEM and DMEM media (Fig 5D and 5E). It was less complex than the lipid A profile in the EMJH and HAN media, with a base peak at *m/z* 1748.26 and additional small lipid A cluster around the *m/z* 1722.25 ion. To confirm that the observed ions were in fact lipid A molecules and not host lipids, the *m/z* 1748.26 ion was analyzed by on-tissue tandem spectrometry. The following diagnostic lipid A ions were identified: *m/z* 110.98 (representing a methylated phosphate), *m/z* 723.46 ($B_1$ ion resulting from fragmentation of the di-aminoglucose backbone), and *m/z* 1299.91 and 1524.09 (lipid A fragment ions resulting from the loss of 2'ε and 3'ε and 2'ε or 3' ε fatty acyl chains, respectively) [60], confirming *m/z* 1748.26 to be *Leptospira* lipid A (S5A Fig). Infected and control hamster liver tissues four days post-challenge were homogenized for lipid A microextraction [62,63]. Lipid A signal was detected in 2/3 extracts from liver of *Leptospira*-infected animals (S5B and S5C Fig) and was not present in any of the four liver extracts from negative controls (S5D Fig). The fragmentation pattern of the extracted lipid A was identical to that obtained by on-tissue tandem mass spectrometry (S5A Fig), further validating our results. The penta-acylated lipid A species identified in EMEM- and DMEM-leptospires were not detected in tissue by either technique. However, it is possible that the abundance of these lipid A species was below the limit of detection of the technique used. Collectively, these results indicate that the lipid A profile of *L. interrogans* is highly influenced by culturing conditions and media components. Furthermore, the lipid A profile of *Leptospira* cultured in EMEM and DMEM media resembles *Leptospira* lipid A observed in the host, highlighting the potential of both media to be used as surrogates of *in vivo* conditions.

## Discussion

*Leptospira* is an environmentally transmitted spirochete that can survive for weeks in water or soil. Pathogenic species of the *Leptospira* genus have the ability to infect, disseminate, and colonize tissues of a wide range of mammal hosts [12]. Each of these environments has a different nutrition composition, osmolarity, and temperature. Previous studies have

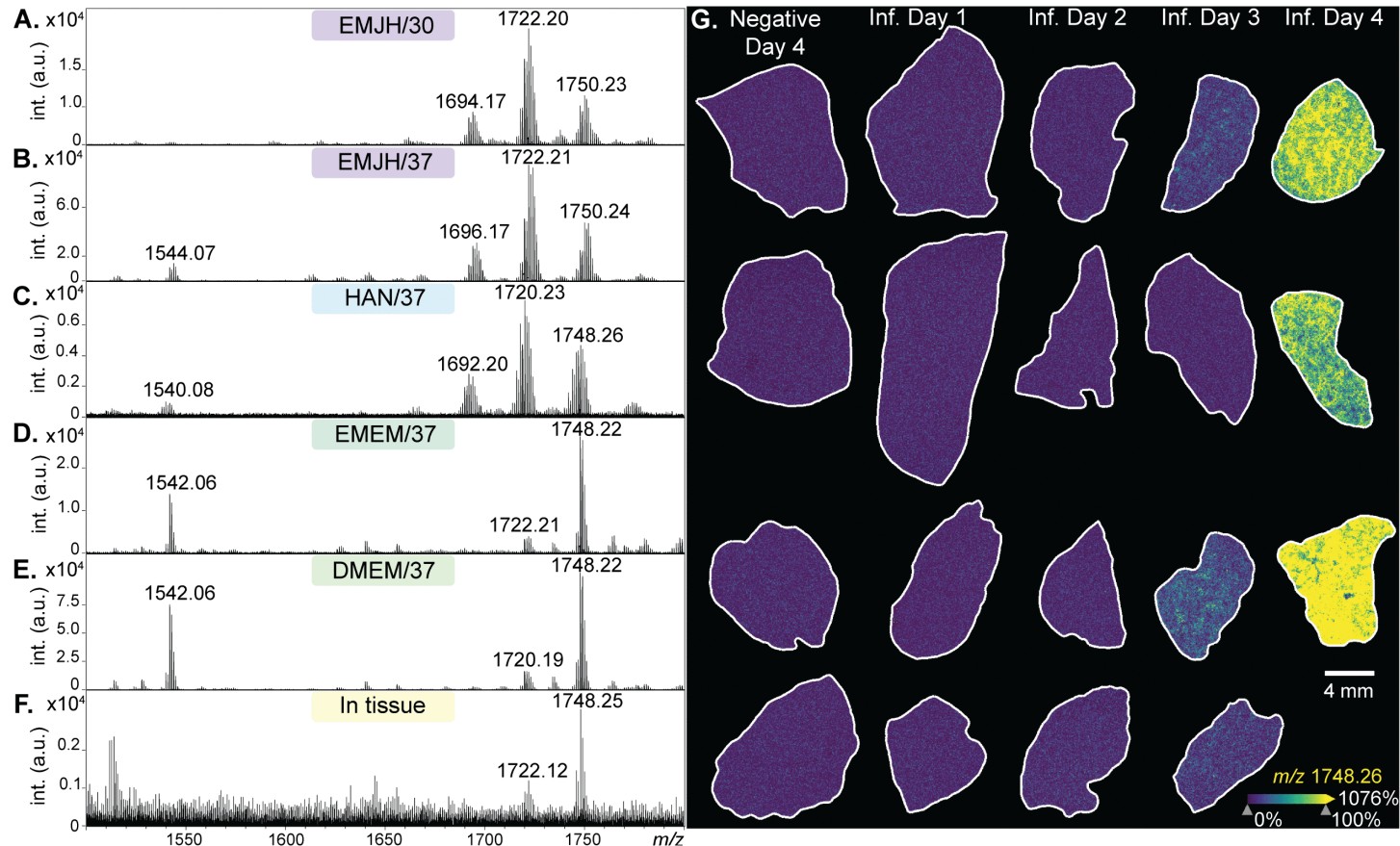

**Fig 5. Lipid A profiles of *L. interrogans* serovar Manilae strain L495 in different growth conditions. A**. EMJH at 30 °C, regular incubator without a $CO_2$ level control. **B**. EMJH at 37 °C and 5% $CO_2$. **C** HAN at 37 °C and 5% $CO_2$. **D**. EMEM at 37 °C and 5% $CO_2$. **E** DMEM at 37 °C and 5% $CO_2$. **(A–E)** Lipid A profiles determined by FLAT. **F**. Liver of an infected hamster. **G**. Ion images of a lipid A ion (*m/z* 1748.26 ± 10 ppm),as acquired in hamster liver tissue. Hamsters were infected with $10^8$ *Leptospira*, and sacrificed at days 1, 2, 3 and 4 (n = 4 per group; one hamster did not survive the challenge). Negative controls were infected with EMJH medium only and sacrificed at day four post-injection (n = 4). Data normalized to the total ion count, 100% on the color scale corresponds to 99% quantile. **(F–G)** Data collected by mass spectrometry imaging at 50 µm spatial resolution. All data acquired in the negative ion mode on a timsTOF flex MALDI-2 instrument (Bruker).

shown that shifting leptospires between different environments and conditions alters the bacterial transcriptome and pro-teome, as the spirochetes adapt to those changes [23,64–70]. Despite recent efforts, the molecular mechanisms behind the pathogenesis and adaptation ability of leptospires to survive on this wide spectrum of environments are still unknown [71]. The lack of an *in vitro* media that can mimic host conditions has been a major barrier and a contributing factor to this knowledge gap.

In this study, temperature and media composition impacted the growth of pathogenic *Leptospira* without changing their pathogenicity. Leptospires are fastidious bacteria that require long chain fatty acids and ammonium ions as source of carbon and nitrogen, respectively, and the presence of select vitamins and nutritional supplements for optimal growth [21]. The most common medium used for *in vitro* culture and propagation of leptospires is the EMJH [20]. Although EMJH offers the bacteria all the nutrients necessary for survival and multiplication, it does not mimic host characteristics and conditions [46], leading to adaptation and potentially attenuation after high passages during *in vitro* culturing [72]. More-over, the different recipes and variations of EMJH available create additional challenges with reproducibility of *in vitro*

experiments [20,73]. Recently, the HAN (Hornsby-Alt-Nally) medium has been proposed to culture pathogenic leptospires directly from host tissues [27]. Even though the components of the HAN medium are chemically defined, the major difference between HAN and EMJH is the source of iron and the addition of DMEM medium. Although the HAN media allowed the leptospires to thrive at 37 °C and 5% CO2, we observed a growth pattern very similar to EMJH at 29 °C, with cell growth peaking at day 10. In contrast, DMEM and EMEM media induced leptospires to a faster exponential growth curve, reaching their peak growth in four days, with similar cell numbers as the maximum peak observed in the HAN medium, but lower than in the EMJH medium. EMEM and DMEM use glucose as source of carbon, with the latter having a higher concentration of glucose, vitamins, and amino acids. Nevertheless, the ability of the *Leptospira* strain Fiocruz L1-130 to cause disease in the hamster model was not affected by the medium type, temperature, or $CO_2$ levels used for *in vitro* cultivation.

The transcriptome of virulent leptospires was highly affected under *in vitro* media that mimic the host environment, inducing gene expressions closely related to *in vivo* conditions. Previous studies have shown that individual conditions, similar to host environment, like osmolarity [26,66,67] or temperature shifts [24,28–30,69,72,74] can alter leptospiral gene and protein expression. Our results showed that exposing leptospires to a host-like environment based on temperature, osmolarity, and levels of $CO_2$, elicited a dramatic change of the leptospiral transcriptome in comparison with the traditional EMJH medium. Furthermore, our results indicated that the transcriptome shifts induced by *in vitro* host-like environment is similar to the shifts observed during the host infection. Previous studies have suggested that vitamin B12 (cobalamin) synthesis is critical for *in vivo* survival and dissemination of pathogenic leptospires, which is highlighted by its absence in saprophytic species of *Leptospira* [48,75]. In our results, we identified a six-gene cluster involved in cob(II)yrinate a,c diamide biosynthesis I (early cobalt insertion; part of the cobalamin biosynthesis pathway) significantly activated in EMEM, DMEM, HAN, and WB samples compared to EMJH, which is highly supplemented with vitamin B12. In contrast, we observed that fatty acid and lipid degradation pathways were downregulated in WB, DMEM, and EMEM only, when compared to EMJH. This pathway is involved in the metabolism of fatty acids and lipids as a source of nutrients and energy for the cell. The upregulation of this pathway in EMJH and HAN media might be related to their high concentration of Tween, used as an energy source. When evaluating the expression of virulence factors, we observed that important genes encoding leptospiral immunoglobulin-like proteins (*ligA* and *ligB*) [26], sphingomyelinase hemolysin (*sph2*) [38,66,76], and collagenase (*colA*) [77,78] where all upregulated on EMEM, DMEM, HAN, and WB samples compared to EMJH. Finally, the gene *lipL36*, which encodes a lipoprotein (LipL36) expressed when *Leptospira* grows *in vitro* at 30 °C but not *in vitro* at 37°C or *in vivo* [79], was downregulated in WB, HAN, EMEM, and DMEM.

The tissue culture media DMEM and EMEM are promising candidates as *in vitro* surrogates for *in vivo* experiments that evaluate leptospiral gene transcription and pathogenesis. Heavy animal use in research has been raising ethical concerns. Leptospiral studies frequently rely on laboratory animal models, even though there is no consensus correlating experimental infection studies with natural infection [7]. A recent study used intraperitoneal dialysis membrane chambers (DMC) that replicated conditions of a natural leptospirosis infection. The authors identified several essential genes modulated *in vivo*, but the method involves animal experiments and surviving surgery [64]. In our experiments, analyzing the transcriptome of hamster blood infected with leptospires, we identified more than 500 differentially expressed genes (DEGs) that do not appear in the DMC approach. When considering the number of DEGs, the correlation of modulated genes, and genes orthologous classification, our analysis showed that DMEM and EMEM media are comparable to WB samples, and thus to *in vivo* conditions. DMEM and EMEM conditions lead to a gene transcription that shared 40% and 47%, respectively, of all differentially regulated genes on the host (WB). In contrast, HAN and DMC shared only 18% and 9%, respectively. Most pathogenesis studies focus on host-pathogen interactions with different animal models, resulting in many animals experiencing painful disease progression. Each animal has a variable reaction to infectious agents based on genetic and environmental conditions and reproducibility is required as the foundation of conducting the experiment [80]. Repeating animal experiments always involves extra cost, time, and other resources. Compassion burnout in

technicians can also present a challenge to proper animal care [81]. Thus, minimizing or reducing animal use and meeting animal welfare requirements are desirable and regulated under the federal government.

The growth media composition and *in vitro* culturing conditions also impacted the *Leptospira* lipid A profile. Many bacteria species, including *Leptospira*, maintain the integrity of their membranes by modifying lipid A structure in response to environmental cues (*e.g.,* temperature shift and presence of antimicrobials) [43,82,83]. Growth in DMEM and EMEM media reduced heterogeneity of *Leptospira* lipid A and induced incorporation of longer secondary fatty acyl chains and the occurrence of penta-acylated lipid A species. *Leptospira* cannot synthesize fatty acids *de novo* and has only a limited capacity to modify them [84]. Therefore, one possible explanation of the observed lipid A diversity is a variation in the availability of the individual fatty acids in different media. Both HAN and EMJH use Tween 80 as a major source of fatty acids (50–70% oleic acid, balanced with linoleic, palmitic, and stearic acids). EMEM and DMEM source of fatty acids is exclusively from the addition of rabbit serum, which overall is composed of palmitic acid (~40%), linoleic acid derivatives (~20%), and oleic acid (~20%) [85]. Nevertheless, changes in the abundances of transcripts of genes involved in lipid A biosynthesis were detected between the growth conditions, suggesting that modifications to lipid A were also regulated on the gene transcription level. Further studies to better understand the effect of the source of fatty acids on the leptospiral lipid A profile should be considered.

EMEM- and DMEM-grown *Leptospira* induced lipid A profile more closely to the bacteria found *in vivo*. Detection of lipid A by MALDI mass spectrometry requires mild-acid hydrolysis that disrupts the covalent bond between the molecule and the rest of the LPS; lipid A otherwise does not ionize [57]. Curiously, *Leptospira* lipid A was readily detectable in the liver of infected hamsters without the need for on-tissue derivatization that ordinarily facilitates the hydrolysis [86]. The direct detection suggests that a portion of lipid A is present in the *Leptospira* membrane in a free form, not bound to LPS. The only other bacterium described to date with free lipid A is *Francisella*. *Francisella* lipid A is also detectable in tissue without needing of acid hydrolysis and its LPS has similar biochemical properties to *Leptospira* LPS [63,87,88]. Lipid A activates the innate immune cascades by binding to the Toll-like receptor 4-myeloid differentiation protein 2 complex (TLR4-MD2). Variation in the number, length, and saturation of lipid A acyl chains affects the binding efficiency, modifying the downstream TLR-induced signalling and the recruitment of immune responses [89–95]. Overall, the *Leptospira* lipid A profile in tissue differed from that of the EMJH-grown *Leptospira* traditionally used to study host immune responses and to generate bacterins for animal vaccines against leptospirosis. In the future, experiments to determine activation of TLR4-MD2 by individual lipid A extracts will be essential to better understand media composition and lipid A structure on immune responses against leptospirosis that can potentially influence current and new vaccine candidate's efficacy studies.

In summary, in this work, we characterize the *in vitro* growth of virulent *Leptospira* sp. using different media and conditions commonly used in various mammalian cell types. Our growth analysis indicated that EMEM or DMEM are good alternative media to grow leptospires in a host-like environment and circumvent the issue of fastidiousness when culturing pathogenic *Leptospira*. After the identification of different profile signatures obtained in each medium, we also observed that commercial media DMEM and EMEM elicit similar leptospiral gene expressions as the *in vivo* environment represented here by whole blood of infected hamsters. Furthermore, *Leptospira* lipid A was, for the first time, characterized directly from the infected tissue and our results showed that DMEM and EMEM media induces a lipid A profile that is more closely related to that *in vivo* when compared to traditional growth conditions. Although the HAN media has been developed and used as an alternative for EMJH to isolate leptospires, HAN is closer to EMJH than the other media used here when considering the induced transcriptomic analysis and lipid A structure. Our results are limited by using a minimal number of replicates (n = 2). Therefore, we used conservative cut-off values for our analysis and validated our findings using additional methods and approaches. We also identified a large number of DEGs only in WB conditions, highlighting the limitations of *in vitro* conditions to mimic the full complexity of the host-pathogen interaction but also providing us with a list of targets that are potentially essential to processes occurring during host infection. Our choice for host-condition sample (7-days post-infection blood) has established biological relevance. However, we do not have a clear

understanding of the transcriptomic profile of leptospires in different tissues and times of the infection process and how can this influence dissemination and disease progression. Furthermore, in this study we didn't evaluate representative strains from all 41 species of *Leptospira* and the feasibility to use EMEM and DMEM on standard laboratory techniques for leptospirosis research, including serological assays. Additional experiments are in progress to address those differences and questions.

Recently, the leptospiral field has seen an increased on the number of novel *Leptospira* species identified [96,97], coupled with an emergence of cases around the world, including in developed countries [98]. In parallel, there have been major advances in development of genetic tools to manipulate leptospires cells [99] and important discoveries on *Leptospira* biology and pathogenicity [71]. Both DMEM and EMEM media can be found worldwide at lower costs compared to EMJH and HAN and have a defined composition eliminating the necessity of in-house media production and variability, an important issue for leptospirosis and the spirochetal research field. The prospect of a commercially available *in vitro* media that can be used as a potential surrogate for the host environment is critical and essential to reduce animal use, simplify the culture of leptospires, and facilitate research, thus allowing for new discoveries on this important neglected disease.

## Methods

### Ethics statement

The protocol of animal experimentation was prepared and approved according to the guidelines of the Institutional Committee for the Use of Experimental Animals at Yale University (protocol # 2023–11424) and Institute Pasteur (protocol # 220016). All animal procedures carried out at Institute Pasteur were performed in accordance with the European Union legislation for the protection of animals used for scientific purposes (Directive 2010/63/EU). Animals were monitored twice daily for endpoints, including signs of disease and death, up to 21-days post-infection. Surviving animals at the end of the experiment (7-days post-challenge) or moribund animals showing difficulty in moving, breathing, and/or signs of bleeding or seizure were immediately sacrificed by inhalation of $CO_2$.

### Bacteria culture, media, and growth conditions

A low-passage of *Leptospira interrogans* serovar Copenhageni strain Fiocruz L1-130 was cultured in four different media (EMJH, EMEM, DMEM, and HAN). We used Ellinghausen–McCullough–Johnson–Harris (EMJH) [18] supplemented with 1% rabbit serum (Sigma-Aldrich) at 30 °C in a shaking incubator (100 rpm). EMEM medium was prepared from Eagle's Minimum Essential Medium (EMEM; ATCC 30–2003) with 5% rabbit serum and 1.8 mM $FeSO_4$. DMEM medium was prepared from Dulbecco's modified Eagle medium (DMEM) supplemented with 5% rabbit serum and 100 μl 0.036 mM FeSO4, and HAN medium was prepared following the methods previously described by the authors [27]. The list of components and total concentration for each media is listed on S11 Table.

Spirochetes were enumerated by dark-field microscopy in a Petroff-Hausser chamber (Thermo Fisher Scientific). Each media was inoculated with $10^4$ of *L. interrogans* serovar Copenhageni strain L1-130 and incubated at 37 °C under 5% $CO_2$ (EMJH 37 °C, EMEM, DMEM, and HAN) and at 29 °C (EMJH 29 °C). The growth curve was evaluated by counting the cultured cells every other day for 14–16 days. The experiment was repeated twice for reproducibility.

### Animal challenge

Golden Syrian Hamsters (*Mesocricetus auratus*) were used as the animal model. Hamsters are highly susceptible to leptospirosis, are the model of choice for acute leptospirosis, and emulate the natural history and clinical presentation of severe leptospirosis in humans [100,101]. Two groups of six 12-week-old male hamsters were infected with a dose of $10^8$ *L. interrogans* serovar Copenhageni strain Fiocruz L1-130 by conjunctival route [35]. After 7-days post-challenge, five hamsters were euthanized, and one hamster was left for confirmation of virulence. During the euthanasia, 3 mL of blood was taken by cardiac puncture from each animal for DNA and RNA extraction. Blood was

collected with and without EDTA for DNA extraction to obtain sera and plasma. For RNA extraction, blood was collected using only EDTA tubes.

For the virulence studies, four groups of three 12-week-old male hamsters were infected with a dose of $10^8$ *L. interrogans* serovar Copenhageni strain Fiocruz L1-130 by conjunctival route as described above. Each group was challenged with leptospires cultured under different media and conditions: HAN, EMEM, and DMEM culture at 37 °C under 5% $CO_2$, and EMJH cultured aerobically at 29 °C as described above.

For the mass spectrometry imaging experiment, four groups of four 12-week-old male hamsters were inoculated intraperitoneally with $10^8$ of low-passage *L. interrogans* serovar Manilae strain L495 grown in EMJH medium to mid-logarithmic phase. Hamsters were sacrificed at Day 1, 2, 3 and 4 post-infections (n = 4 per group). One out of sixteen hamsters succumbed to infection before the experimental end point, and was, therefore, not included in the analysis. Control group (n = 4) was inoculated intraperitoneally with EMJH medium only and sacrificed at Day 4 post-injection. Liver was collected and immediately frozen by floating in vapors of liquid nitrogen. Tissues were stored protected from moisture at −80°C until processing.

## DNA and RNA extraction

Sera and plasma were obtained by centrifugation of clotted blood and whole blood with EDTA at 1,000 × g for 15 min at room temperature. DNA was extracted from 250 µL of hamsters' serum, plasma, and whole blood using Maxwell 16 tissue DNA purification kit (Promega), following the manufacturer's instructions and using a 200-elution volume.

We extracted RNA for two biological replicates of each culture condition and hamster blood. A volume of 20 mL of leptospires cultured on different media at late-logarithmic phase and EDTA tubes containing ~3 mL of whole blood from each hamster were centrifuged at 6000 x g for 15 minutes. Trizol (Invitrogen) was added to the pellets for RNA stabilization (3:1 *v/v*). RNA isolation was performed on 400 µL of the mix using Direct-zol RNA MiniPrep (Zymo Research), following manufacturer's instructions. TURBO DNA-free kit (Invitrogen) was used to eliminate DNA residuals. RNA concentration, purity, and quality were verified using Qubit and NanoDrop instruments (Thermo Fisher Scientific). Samples were stored at −80°C until downstream analysis.

## RNA-sequencing and data analysis

Samples (n = 2) for all our conditions were sent to Yale Center for Genome Analysis (YCGA) for bioanalyzer, to check integrity and concentration. Only samples with RIN > 7 were chosen for mRNA library construction using TruSeq standard mRNA Library Prep kit (Illumina) and sequenced on NovaSeq 6000 (Illumina) with 100 base pair paired end reads. Prior to sequencing on the Illumina NovaSeq platform, bacterial ribosomal RNA was depleted using the Illumina Ribo-Zero kit. To increase the RNA alignment quantity from *Leptospira* extracted from bacterial cells using blood of hamsters, we supplied multi-targeted primers (MTPs) [102] to the reverse transcription to specifically generate cDNA from *Leptospira* genomic mRNA: degenerated primers were sorted and scored for their binding preferences against the genome of *Leptospira* strain Fiocruz L1-130 and the host hamster. The primer pool with the highest MTP score was used for the cDNA libraries preparation. Trimmed raw reads were aligned to the reference genome of *L. interrogans* serovar Copenhageni strain Fiocruz L1-130 from NCBI (267671) using HISAT2 v2.1 [103]. Alignments with a quality score below 20 were excluded from further analysis. Reads were counted for each gene with StringTie v1.3.3 [103], and differentially expression analysis was performed in DESeq2 [104]. *P*-values were corrected for multiple testing using the Benjamini-Hochberg method. Adjusted *P*-values (Padj) < 0.01 and log2 fold change (log2FC) ± 2 were used as the criteria to determine significant differences in gene expression. PCA plots showed samples clustering by different culture medium. Genes regulating virulence factors or regulatory networks in leptospirosis were selected based on previous literature [42,47–51] for further analysis.

## qPCR and RT-qPCR for target gene identification and RNA seq data validation

Using the extracted DNA, the concentration of leptospires was quantified by a TaqMan-based quantitative-PCR assay using an ABI 7500 system (Thermo Fisher Scientific) and Platinum Quantitative PCR SuperMix-UDG (Thermo Fisher Scientific). The qPCR reaction was performed using *lipL32* primers and probes as previously described [34], with addition of a Xeno internal positive control (Applied Biosystems) to verify inhibition. Based on a standard curve, the bacterial quantification was calculated and expressed as the number of leptospires per milliliter. For the RT-PCR, the High-capacity cDNA reverse transcription kit (Applied Biosystems) was employed to convert total RNA to single-stranded cDNA. The RT-PCR was performed on 7500 fast real-time PCR (Applied Biosystems) using iQTM SYBRR Green Super-Mix (Bio-Rad) according to the manufacturer's instructions. For the endogenous control, we used the 23S rRNA gene. The thermal cycling conditions were 95°C for 3 min, followed by 40 cycles of 95°C for 5 s and 60°C for 1 min. The specificity of the SYBR green PCR signal was confirmed by melt curve analysis. A relative quantification analysis was performed using the comparative CT method, and the relative gene expression was calculated by using the $2 - \Delta\Delta Ct$ method [105]. Serial dilutions of the cDNA were performed to verify the absence of inhibition on our assays.

## EGGNOG classification and enrichment of biological pathways

We assessed the cluster of orthologous genes (COG) classification system to extract important information from our gene lists according to their homologous relationships [106]. We use the Pathway tool from BioCyc to analyze biological pathways with an enriched list of differentially expressed genes between treatments [52].

## *Leptospira* lipid A extraction

**Cultured cells.** *L. interrogans* serovars Manilae strain L495 [107] and serovar Copenhageni strain Fiocruz L1-130 were grown in triplicates to a mid-logarithmic growth phase (approximate density of $5 \times 10^8$ cells/mL; as determined by dark field microscopy). Five milliliters of culture were pelleted *via* centrifugation at $4,000 \times g$ for 15 minutes. Pellets were washed twice with 1 mL of phosphate buffered saline (Sigma Aldrich) and reconstituted with 200 μL of MS-grade water (Fisher Chemical). Lipid A was isolated from the resulting cell suspension by the Fast Lipid Analysis Technique (FLAT$^n$) [57,58]. Briefly, one microliter of the sample was spotted in triplicate on a disposable Matrix-Assisted Laser Desorption/Ionization (MALDI) plate (MFX μFocus plate 12 × 8 c 2,600 μm, Hudson Surface Technology) and dried. One microliter of citric acid buffer (0.2M citric acid, 0.1M trisodium citrate, Fisher Chemical) was then deposited over the samples. Plates were incubated over in a glass humidifying chamber filled with 4 mL of MS grade water at 110 °C for 30 minutes, allowed to cool, and washed with ~10 − 15 mL deionized water.

**Infected tissues.** Lipid A was extracted from tissues according to the lipid A microextraction protocol with modifications [62,63]. Briefly, approximately 50 mg of *Leptospira*-infected or uninfected (control) hamster liver tissue was homogenized using a Retsch MM 400 homogenizer (Retsch) at 28Hz for 2 × 2min. Samples were cooled in liquid nitrogen vapor between the cycles. The homogenized tissue was resuspended in 1.2 mL of isobutyric acid: 1 M ammonium hydroxide (5:3, *v/v*; Sigma Aldrich) and incubated for 2 h at 100 °C with occasional vortexing. Samples were cooled on ice and centrifuged at $2000 \times g$ for 15 min at room temperature. Pellets were washed twice with methanol. The methanol-insoluble lipid A was then extracted from the pellet using chloroform: methanol: water (3: 1.5: 0.25, *v/v*; Fisher Chemical). Samples were centrifuged for 1.5 min at $6000 \times g$, and lipid A was re-extracted with chloroform: methanol (2:1, *v/v*). Both lipid A extracts were combined in an autosampler vial with a glass insert and brought to dryness under a stream of heated nitrogen gas (35 ºC). Finally, samples were rehydrated in 10 μL of chloroform: methanol (2:1, *v/v*), and 1.5 μL was spotted on a disposable MALDI plate.

## Mass spectrometry analysis of extracted *Leptospira* lipid A

Lipid A extracts were overlaid with one microliter of norharmane matrix (Sigma Aldrich) dissolved to 10mg/mL in MS grade chloroform: methanol (2:1, *v/v*). Data was acquired on a timsTOF flex MALDI-2 (Bruker) in the negative ion

mode with 1 burst of 1,000 shots per spot at 5,000 Hz. The following settings were used: scan range $m/z$ $600-2300$, collision RF 4,000 Vpp, pre-pulse storage 15.0 ms, and transfer time 150 ms. Lipid A ions were analyzed by tandem mass spectrometry, using a 1,000 shots per spot, $m/z$ 4 isolation width and variable collision energy ($100-120$V), collision RF ($700-4,000$ Vpp), pre-pulse storage ($7.0-15.0$ ms), and transfer time ($70-150$ ms) to generate a full mass fragment signature. Mass spectra were analyzed using Compass Data Analysis v6.0 (Bruker) and mMass v5.5.0 [108].

## Mass spectrometry imaging

Liver tissues were sectioned to 10 µm thickness on a C1950 cryostat (Leica Biosystems); chamber temperature was set to −17 °C, specimen head cooling was turned off. Sections were thaw-mounted on indium tin oxide slides (Delta Technologies), dried in a nitrogen box for 30 minutes and stored protected from moisture at −80 °C until processing. Samples were taken out of the freezer in randomized order, thawed sealed at room temperature, and dried in a nitrogen box for 30 minutes. One microliter of SPLASH LIPIDOMIX Mass Spec Standard (Croda) in MS-grade methanol (Fisher Chemical) was deposited on the slide separated from the tissue for quality control purposes. Norharmane matrix (7 mg/ml; Sigma Aldrich) in 2:1 (v/v) MS-grade chloroform and methanol (Fisher Chemical) was deposited on tissue sections using an automated pneumatic HTX M5 sprayer (HTX Technologies). Spraying conditions were as follows: 12 passes with 30 s drying time between them, 30 °C nozzle temperature, 0.12 ml/min flow rate, 1200 mm/min velocity, 3 mm track spacing, crisscross pattern, 10 psi pressure, 2 l/min gas flow rate, 40 mm nozzle height and heated tray temperature at 25 °C [109]. Slides were sprayed in batches of three and kept in the nitrogen box until processing. Optical images were acquired using an office scanner (Epson) and regions for imaging were outlined in Flex Imaging (Bruker): tissue, matrix only (negative control), and an area with the SPLASH LIPIDOMIX standard (quality control). A timsTOF flex MALDI-2 instrument was calibrated in the electrospray mode using a direct infusion of the Agilent Tuning Mix (Agilent Technologies). Mass spectra were acquired using the following settings: custom laser with a single smart beam ($46\times46$ µm; resulting in 50µm pixel size), scan range $m/z$ $600-2300$, 250 shots per spot at 5,000 Hz, collision RF 4,000 Vpp, pre-pulse storage 15.0 ms, and transfer time 150 ms. The following ions were used for online calibration: $m/z$ 515.1620 (norharmane matrix ion) $m/z$ 885.5497 (phosphatidylinositol, PI 38:4), $m/z$ 1447.9650 (cardiolipin, CL 72:8) and $m/z$ 1449.9806 (cardiolipin, CL 72:7). On-tissue tandem mass spectrometry analysis of lipid A was performed using the same parameters as described above for on-plate tandem mass spectrometry. Raw data were processed in SCiLS Lab v2024a Pro (Bruker) and ion images were created upon normalization to total ion count.

## Statistical analysis

GraphPad Prism (Prism Mac 5.0) was used to analyze *in vivo* and *in vitro* assays statistically. Real-time qRT-PCR data were analyzed using one-way ANOVA with Bonferroni post hoc test at $P<0.05$. Benjamini-Hochberg FDR was use as cut off for RNA seq data. Deseq2 Package was used to perform differential expressed analysis [104]. Gene transcription data was normalized, log-transformed, and Pearson correlation was used to compare RNA expression between different treatments.

## Supporting information

**S1 Fig. Results of qPCR for *Leptospira* using DNA extracted from whole blood, serum, and plasma of hamsters after 7 days post-conjunctival infection with $10^8$ leptospires (*L. interrogans* serovar Copenhageni strain Fiocruz L1-130).** Numbers are expressed in logarithmic genome equivalent (GEq) with average and standard deviation. *: $P<0.05$.
(TIF)

                                                                    

**S2 Fig. Relative expression of all the genes involved in Cobalamin production in different *in vitro* growth conditions and in whole blood (WB).** Bars represent expression levels of each gene on leptospires cultured in different media. (TIF)

**S3 Fig. Pearson correlation coefficient results for validation of RNA-Seq analysis using RT-qPCR.** Dots represent comparison results of four genes tested on all four media conditions analyzed: EMJH, HAN, EMEM, and EMEM. Data from RNAseq is plotted on Y-axis and qRT-PCR is plotted on y-axis with columns representing the level of expression for each analysis. (TIF)

**S4 Fig. Comparison of Lipid A profiles of *L. interrogans* serovar Manilae strain L495 and serovar Copenhageni strain Fiocruz L1-130 in different growth conditions.** Strains L495 (M) and Fiocruz L1-130 (C) were cultured in HAN and EMEM and lipid A profile was analyzed by mass spectrometry. (TIF)

**S5 Fig. Mass spectrometry analysis of *Leptospira* lipid A from infected hamster liver.** A. Tandem mass spectrometry analysis of the *m/z* 1748.26 lipid A ion. Top panel (purple): lipid A in tissue by mass spectrometry imaging. Bottom panel (green): lipid A extracted from liver homogenates. **B–C.** Lipid A signal in liver extracts from hamster four days post-challenged with *L. interrogans* serovar Manilae strain L495. Tissue used for extraction correspond to Fig 4G in the main manuscript (Day 4, tissue section on the bottom and top of the panel; lipid A extraction from the tissue depicted in the middle was not successful). **D.** Representative spectrum of liver extracts from uninfected controls; all negative for presence of the lipid A signal (n = 4). Tissues used for extraction correspond to Fig 5G in the main manuscript (Negative, Day 4). (TIF)

**S1 Table. List of DEGs identified between all different media evaluated when compared to EMJH.** (XLSX)

**S2 Table. Top 10 up and downregulated leptospiral genes identified in whole blood (WB) compared to DMEM, EMEM, and HAN versus EMJH. p-value<0.01 and log2FC±2.** (DOCX)

**S3 Table. Biological pathways identified with genes differentially expressed between different growth conditions and environments compared to EMJH (*P*<0.05).** (DOCX)

**S4 Table. Biological pathways identified with genes differentially expressed between DMEM and EMJH media.** (DOCX)

**S5 Table. Biological pathways identified with differentially expressed genes between EMEM and EMJH media.** (DOCX)

**S6 Table. Biological pathways identified with differentially expressed genes between HAN and EMJH media.** (DOCX)

**S7 Table. Biological pathways identified with differentially expressed genes between WB and EMJH.** (DOCX)

**S8 Table. Percentual of differently expressed genes between different media conditions and dialysis membrane chamber (DMC) compared to whole blood (WB) samples.** (DOCX)

**S9 Table. List of up- and down-regulated genes identified in whole-blood only when compared to EMJH.** (XLSX)

**S10 Table. Differential expression of genes related to lipid A biosynthesis in each medium compared to EMJH.** (DOCX)

**S11 Table. List of components and concentrations for each media.** (XLSX)

## Author contributions

**Conceptualization:** Helena Pětrošová, Elsio A Wunder Jr.

**Data curation:** Leandro E Garcia, Zitong Lin, Zheng Wang, Francesc Lopez-Giraldez, Helena Pětrošová, Elsio A Wunder Jr.

**Formal analysis:** Leandro E Garcia, Zitong Lin, Sophie Culos, Zheng Wang, Francesc Lopez-Giraldez, Angela M. Jackson, Helena Pětrošová, Elsio A Wunder Jr.

**Funding acquisition:** David R. Goodlett, Jeffrey P Townsend, Helena Pětrošová, Elsio A Wunder Jr.

**Investigation:** Leandro E Garcia, Zitong Lin, Sophie Culos, M Catherine Muenker, Emily E. Johnson, Angela M. Jackson, Elsio A Wunder Jr.

**Methodology:** Leandro E Garcia, Angela M. Jackson, Helena Pětrošová, Elsio A Wunder Jr.

**Project administration:** Helena Pětrošová, Elsio A Wunder Jr.

**Resources:** Alexandre Giraud-Gatineau, Mathieu Picardeau, David R. Goodlett.

**Supervision:** David R. Goodlett, Jeffrey P Townsend, Helena Pětrošová, Elsio A Wunder Jr.

**Validation:** Leandro E Garcia, Helena Pětrošová, Elsio A Wunder Jr.

**Visualization:** Leandro E Garcia, Sophie Culos, Helena Pětrošová, Elsio A Wunder Jr.

**Writing – original draft:** Leandro E Garcia, Zitong Lin, Helena Pětrošová, Elsio A Wunder Jr.

**Writing – review & editing:** Leandro E Garcia, Sophie Culos, Zheng Wang, Francesc Lopez-Giraldez, Alexandre Giraud-Gatineau, Angela M. Jackson, Mathieu Picardeau, Jeffrey P Townsend, Helena Pětrošová, Elsio A Wunder Jr.

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
