## [Decision Letter · Decision Letter 0]

23 Oct 2025

PNTD-D-25-01540

DMEM and EMEM are suitable surrogate media to mimic host environment and expand leptospiral pathogenesis studies using in vitro tools

Dear Dr. Wunder Jr,

Thank you for submitting your manuscript to PLOS Neglected Tropical Diseases. After careful consideration, we feel that it has merit but does not fully meet PLOS Neglected Tropical Diseases's publication criteria as it currently stands. Therefore, we invite you to submit a revised version of the manuscript that addresses the points raised during the review process.

Please submit your revised manuscript within 60 days Dec 22 2025 11:59PM. If you will need more time than this to complete your revisions, please reply to this message or contact the journal office at plosntds@plos.org. Please include the following items when submitting your revised manuscript:

We look forward to receiving your revised manuscript.

Kind regards,

Roberta Olmo Pinheiro, PhD

Section Editor

Roberta Pinheiro

Section Editor

Shaden Kamhawi

co-Editor-in-Chief

Paul Brindley

co-Editor-in-Chief

**Journal Requirements:**

At this stage, the following Authors/Authors require contributions: Leandro E Garcia, Zitong Lin, Sophie Culos, M Catherine Muenker, Emily Johnson, Zheng Wang, Francesc Lopez-Giraldez, Alexandre Giraud-Gatineau, Angela Jackson, Mathieu Picardeau, David Goodlett, Jeffrey P Townsend, Helena Petrosová, and Elsio A Wunder Jr. Please ensure that the full contributions of each author are acknowledged in the "Add/Edit/Remove Authors" section of our submission form.

4) We do not publish any copyright or trademark symbols that usually accompany proprietary names, eg ©,  ®, or TM  (e.g. next to drug or reagent names). Therefore please remove all instances of trademark/copyright symbols throughout the text, including:

- ® on page: 17.

5) Please upload all main figures as separate Figure files in .tif or .eps format. For more information about how to convert and format your figure files please see our guidelines:

6) We have noticed that you have uploaded Supporting Information files, but you have not included a list of legends. Please add a full list of legends for your Supporting Information files after the references list.

7) Please amend your detailed Financial Disclosure statement. This is published with the article. It must therefore be completed in full sentences and contain the exact wording you wish to be published.

**Reviewers' Comments:**

Reviewer's Responses to Questions

**Key Review Criteria Required for Acceptance?**

**Methods:**

-Are the objectives of the study clearly articulated with a clear testable hypothesis stated?

-Is the study design appropriate to address the stated objectives?

-Is the population clearly described and appropriate for the hypothesis being tested?

-Is the sample size sufficient to ensure adequate power to address the hypothesis being tested?

-Were correct statistical analysis used to support conclusions?

-Are there concerns about ethical or regulatory requirements being met?

Reviewer #1: yes

Reviewer #2: The manuscript is clearly written, and the goal, hypotheses and design of the study are clearly written. I don't think there is any ethical and regulatory problem in the study.

Specific comments

In this study, only two strains, strain Fiocruz L1-130 and strain L495, were used in the experiments. However, there are two concerns related to this. First, the strain used in the mass spectrometry analysis (L495) is different from the strain used in the other analyses (L1-130). Is it really possible to link the results across different types of analyses despite the different strains used? Second, in each analysis, only one strain (L1-130 or L495) was tested. Is it really possible to generalize the results from one strain to other strains? Was there a need to investigate variation across strains? Please mention these issues.

Line168

Why were the measurements taken place at the late-logarithmic stage? Does this in vitro stage truly correspond to the timing of WB collected from hamsters in vivo (i.e., 7 days from the start of challenge)?

Lines 203 and 206 (and Fig. 3A)

The Pearson correlation coefficient is a parametric method that assumes that the data for both variables are normally distributed. Is the normality of the variables checked? If they are not normally distributed, non-parametric methods such as the Spearman rank correlation would be a better choice.

Line 523

Were ribosomal RNAs (rRNAs) removed? The inclusion of rRNAs affects mRNA sequencing depth by RNA-seq.

Line 541

For qPCR and RT-qPCR, did the authors verify that there was no inhibition in PCR? Inhibition of PCR amplification affects quantitative results.

Line 545-546

Can this primer pair really tell the number of bacterial cells? In other words, is the gene targeted by this primer pair single copy per bacterial cell? If not, the results will not directly represent bacterial cell number.

Line 629

Here, the authors used ANOVA, a parametric test. Did the authors check the normality of the results?

Reviewer #3: (No Response)

Reviewer #4: The manuscript explores the use of DMEM and EMEM as alternative culture media to EMJH for Leptospira interrogans, aiming to better mimic host-like conditions and expand the utility of in vitro tools for studying leptospiral pathogenesis. The study is generally well-structured, with clearly described experimental design and methodology. An appropriate ethics statement is also included.

Reviewer #5: (No Response)

**Results**

-Does the analysis presented match the analysis plan?

-Are the results clearly and completely presented?

-Are the figures (Tables, Images) of sufficient quality for clarity?

Reviewer #1: (No Response)

Reviewer #2: The results are reported in an easy-to-understand manner in accordance with the research plan and methodology presented in the manuscript. The figures are generally well prepared and clearly presented. However, some supplementary tables are not found in the manuscript package (see below comments).

Specific comments

Line 179

Were all or part of DEGs associated with virulence mechanisms selected? If only part of DEGs associated with virulence mechanisms were selected, please clarify the selection criteria.

Lines 217-219

The fact written here is difficult to understand just by looking at the Venn diagrams (Fig. 4A), so please explain them in more detail, either by modifying the diagram or by adding more detailed explanations in the text.

Line 222

I cannot find S8 Table in the manuscript package.

Line 255

I cannot find S10 Table in the manuscript package.

Reviewer #3: (No Response)

Reviewer #4: The title suggests a focus on pathogenesis, yet the link between in vitro culture conditions and in vivo pathogenesis remains somewhat ambiguous. While the authors analyze transcriptomic changes and perform in vivo experiments using hamsters, it is unclear how the media-induced transcriptional differences directly influence disease progression or severity in vivo. Clarification on this link would strengthen the conclusions.

EMJH remains the standard medium for culturing Leptospira strains and has proven utility in various applications, including serological assays such as the microscopic agglutination test (MAT). The manuscript raises valid concerns about the limitations of EMJH, particularly regarding potential attenuation of strains, but does not adequately address how media changes might impact downstream applications, including immunological or diagnostic tests.

It is also unclear whether the observed differences in growth kinetics across media types are statistically significant. For example, in Figure 2A, there appears to be a marked difference between EMJH and other media, yet the statistical treatment of this variance is not clearly presented.

Furthermore, the authors do not include a direct comparison with environmental or recently isolated host-derived strains, which would provide valuable context. The use of L. interrogans strain Fiocruz L1-130, while relevant, still represents a laboratory-adapted strain. It would be informative to compare its genetic or phenotypic characteristics to those of true wild-type isolates, especially in relation to culture-induced adaptations.

Reviewer #5: (No Response)

**Conclusions**

-Are the conclusions supported by the data presented?

-Are the limitations of analysis clearly described?

-Do the authors discuss how these data can be helpful to advance our understanding of the topic under study?

-Is public health relevance addressed?

Reviewer #1: this is a descriptive study, and little justification for a number of conclusions and findings is provided. Why focus on Lipid A genes? What not other genes such as O-antigen loci? What is the significance of the cobalamin gene regulation differences observed? What do the authors think about the mechanisms of differences in gene expression between the different media and whole blood incubation? The authors do not explain why less than 50% of the genes seem to be regulated by the DMEM/EMEM compared to whole blood? There is still a gap. Many putative virulence genes regulation

Reviewer #2: The significance and public health relevance of the study are clearly explained. However, limitations of the study, such as the limited number of Leptospira strains used in the experiments and the lack of generalizability of the results to other strains, are not mentioned.

Reviewer #3: (No Response)

Reviewer #4: The findings on lipid A profile changes are intriguing, but the conclusions drawn from these observations appear somewhat overextended based on the current data. The discussion would benefit from a more focused interpretation of the authors’ own results, rather than speculative extrapolation.

Reviewer #5: RNA-Seq of L. interrogans in the two new media and isolated from hamster blood showed less than 50% correlation between the culture media and the blood. It is hyperbole to claim that the new media mimic or can replace results from animal infection. Please revise the title and manuscript to eliminate the exaggerated claims.

L. interrogans colonize numerous tissues in the mammalian body, including kidneys. The authors examined only leptospires in blood. It is inappropriate to describe bacteria in blood as the host environment. Suggest revising the title to "... infection of host blood..."

**Editorial and Data Presentation Modifications?**

Reviewer #1: The primary raw data are presented as uploaded to the short read archives. The authors need to provide a comprehensive list primary gene up/down regulation in Excel spreadsheets before this paper can be accepted. They relegate such analysis to 10 top unregulated and 10 down regulated genes, and only in the supplementary information. SRA data deposition is not visible to this reviewer. There must be a comprehensive list provided as .xls file of all genes, including the raw data of up/down regulation found in the different replicates.

This manuscript is generally well written but number small grammatical and spelling errors are present.

Reviewer #2: Line 185

Please indicate with an * symbol which tiles in the heatmap in Fig. 2C are statistically significant. Please also clearly indicate what the color scale values in Fig. 2C represent.

Line 203

Please explain in detail what each axis of each bar chart in S3 Fig. represents.

Line 206 and others

The use of the word “expression” is inaccurate because this study only investigated “transcription” of mRNAs. In molecular biology, the word “expression” usually refers to the entire process leading to the production of a functional product (e.g., protein). Therefore, in this case, the word “transcription” would be better.

Reviewer #3: (No Response)

Reviewer #4: Title should be changed to reflect the study design

Reviewer #5: The media consist of DMEM or EMEM plus additional ingredients. It is misleading to refer to the media as "DMEM" and "EMEM", since neither consists of just those single ingredients. New names must be given, and used in the title and throughout the manuscript.

**Summary and General Comments:**

Reviewer #1: A confirmatory paper without mechanism and with only a fraction of relevant data to the field is presented.

Reviewer #2: The manuscript is clearly written, and the goal of the study is clear. The methods used in the experiment are mostly reasonable. However, as mentioned above, one thing I would like to emphasize is that in this study, experiments were conducted on only one strain, and it may be difficult to generalize the experimental results obtained from one strain to other strains.

Reviewer #3: This manuscript addresses the significant and long-standing challenge of developing an in vitro culture system that faithfully mimics the mammalian host environment for pathogenic Leptospira. The authors present a well-designed study that employs a multi-omics approach, combining transcriptomics and lipid A profiling, to argue that commercial media DMEM and EMEM are superior surrogates compared to traditional EMJH or the newer HAN medium. The research question is highly relevant, and the findings have the potential to significantly impact the field by providing a more reliable model for pathogenesis studies while reducing the reliance on animal experimentation.

While the study is commendable for its clear objective and new techniques, there are several critical issues in the experimental design and data analysis that must be addressed to substantiate the main conclusions.

Major Comments

Experimental Design: The Confounding Effect of Temperature. The most critical flaw in the current study design is the comparison of Leptospira grown in DMEM, EMEM, and HAN at 37°C against a control group grown in EMJH at 29°C. While the authors acknowledge that growth in EMJH at 37°C was not successful under their specific conditions (low inoculum), the absence of this crucial control group severely undermines the central conclusion. The distinct clustering of the EMJH group in the PCA analysis and the observed differential gene expression could be largely, if not entirely, attributable to the temperature difference rather than the nutritional composition of the medium. The authors' conclusion that DMEM/EMEM are better compositional surrogates is not fully supported without decoupling the effects of temperature. The authors should make a concerted effort to include a 37°C EMJH group, perhaps by modifying the inoculum density as they themselves cited from previous studies. Without this control, the interpretation of the entire transcriptomic dataset remains ambiguous.

The use of only two biological replicates (n=2) for the RNA-seq analysis provides limited statistical power and is below the standard practice for such studies, which typically require a minimum of three replicates to ensure the reliability of differential expression analysis. This limitation should be transparently stated at the beginning of the results section, not just in the discussion. While re-sequencing may not be feasible, the authors can substantially strengthen their claims by expanding the validation of their findings. The current qPCR validation is a good start but is insufficient. It is strongly recommended that the authors perform additional qPCR assays for a larger set of key genes, particularly those involved in the central biological pathways discussed, such as fatty acid metabolism and vitamin B12 biosynthesis, to provide more robust support for the RNA-seq results.

Minor Comments

The four-way Venn diagram presented in Figure 4 is visually complex and hinders the effective interpretation of the overlaps between the differentially expressed gene sets. The authors are encouraged to replace this figure with an UpSet plot.

Comparative Table of Media Composition. To facilitate a clearer understanding of the potential mechanisms driving the observed differences, the manuscript would benefit greatly from a comparative table. This table should detail the key compositional and physicochemical properties of each medium used (EMJH, HAN, DMEM, EMEM), highlighting differences in carbon sources, amino acids, vitamins, and fatty acid sources (e.g., Tween 80 vs. serum). This would provide readers with a direct basis for interpreting the metabolic shifts observed.

The current analysis is informative but relies on predefined DEG lists. Add GSEA to capture coordinated pathway changes that may be missed by DEG thresholds.

Discussion: Include a brief outlook on future work and applications of these findings.

Temperature and typographical consistency Fig. 1A. “29 °C”, “37 °C”.

Reviewer #4: This is a valuable study that proposes a novel approach to culturing Leptospira under more host-relevant conditions. However, the manuscript would be strengthened by:

• Clarifying the link between in vitro findings and in vivo pathogenesis.

• Including statistical analysis of growth data.

• Discussing the potential impact of alternative media on downstream applications.

• Incorporating comparisons with wild-type or environmental isolates.

• Tempering conclusions to reflect the experimental scope.

Reviewer #5: The authors developed two formulations for cultivation of Leptospira interrogans. Both appear to be easy to prepare, and support good growth of a strain of L. interrogans. The authors found that leptospires cutlured in either of the new media produced lipid A and showed global transcript levels that were closer to those of L. interrogans isolated from infected hamster blood. However, the RNA-Seq data showed less than 50% correlation between either media and hamster blood.

The manuscript suffers from excessive hyperbole, with the authors repeatedly claiming that their media "mimic" the host blood environment.

PLOS authors have the option to publish the peer review history of their article (what does this mean? ). If published, this will include your full peer review and any attached files.

**Do you want your identity to be public for this peer review?** For information about this choice, including consent withdrawal, please see our Privacy Policy .

Reviewer #1: No

Reviewer #2: No

Reviewer #3: No

Reviewer #4: **Yes:**  Kalpana Agnihotri

Reviewer #5: No

**Figure resubmission:**
---

## [Decision Letter · Decision Letter 1]

30 Jan 2026

DMEM and EMEM as alternate growth media for pathogenic Leptospira

Dear Dr. Wunder Jr,

Thank you for submitting your manuscript to PLOS Neglected Tropical Diseases. After careful consideration, we feel that it has merit but does not fully meet PLOS Neglected Tropical Diseases's publication criteria as it currently stands. Therefore, we invite you to submit a revised version of the manuscript that addresses the points raised during the review process.

Please submit your revised manuscript within by Mar 01 2026 11:59PM. If you will need more time than this to complete your revisions, please reply to this message or contact the journal office at plosntds@plos.org. Please include the following items when submitting your revised manuscript:

We look forward to receiving your revised manuscript.

Kind regards,

Feng Xue, Ph.D.

Guest Editor

Roberta Pinheiro

Section Editor

Shaden Kamhawi

co-Editor-in-Chief

Paul Brindley

co-Editor-in-Chief

**Reviewers' Comments:**

Reviewer's Responses to Questions

**Key Review Criteria Required for Acceptance?**

**Methods**

-Are the objectives of the study clearly articulated with a clear testable hypothesis stated?

-Is the study design appropriate to address the stated objectives?

-Is the population clearly described and appropriate for the hypothesis being tested?

-Is the sample size sufficient to ensure adequate power to address the hypothesis being tested?

-Were correct statistical analysis used to support conclusions?

-Are there concerns about ethical or regulatory requirements being met?

Reviewer #2: (No Response)

Reviewer #3: (No Response)

Reviewer #5: (No Response)

**Results**

-Does the analysis presented match the analysis plan?

-Are the results clearly and completely presented?

-Are the figures (Tables, Images) of sufficient quality for clarity?

Reviewer #2: (No Response)

Reviewer #3: (No Response)

Reviewer #5: (No Response)

**Conclusions**

-Are the conclusions supported by the data presented?

-Are the limitations of analysis clearly described?

-Do the authors discuss how these data can be helpful to advance our understanding of the topic under study?

-Is public health relevance addressed?

Reviewer #2: (No Response)

Reviewer #3: (No Response)

Reviewer #5: The two new formulations of media do not result in gene expression profiles that "mimic" profiles in blood or other natural tissues/material. The authors' rebuttal comments do not adequately address this point, and this aspect of the manuscript appears to be identical to the original version. Pretending that less than 50% correlation is "mimicking" is beyond hyperbole and approaches fantasy

**Editorial and Data Presentation Modifications?**

Reviewer #2: (No Response)

Reviewer #3: (No Response)

Reviewer #5: (No Response)

**Summary and General Comments**

Reviewer #2: The authors have responded appropriately to my comments, and I would like to thank them for their sincere consideration of my comments.

Reviewer #3: This revised version has answered the questions I raised very well， I have no further concerns.

Reviewer #5: (No Response)

PLOS authors have the option to publish the peer review history of their article (what does this mean? ). If published, this will include your full peer review and any attached files.

**Do you want your identity to be public for this peer review?** For information about this choice, including consent withdrawal, please see our Privacy Policy .

Reviewer #2: No

Reviewer #3: **Yes:**  Zhe Wang

Reviewer #5: No

**Figure resubmission:**
---

## [Decision Letter · Decision Letter 2]

10 Mar 2026

Dear Dr. Wunder Jr,

We are pleased to inform you that your manuscript 'DMEM and EMEM as alternate growth media for pathogenic Leptospira' has been provisionally accepted for publication in PLOS Neglected Tropical Diseases.

Best regards,

Feng Xue, Ph.D.

Guest Editor

Roberta Pinheiro

Section Editor

Shaden Kamhawi

co-Editor-in-Chief

Paul Brindley

co-Editor-in-Chief

Reviewer's Responses to Questions

**Key Review Criteria Required for Acceptance?**

**Methods**

-Are the objectives of the study clearly articulated with a clear testable hypothesis stated?

-Is the study design appropriate to address the stated objectives?

-Is the population clearly described and appropriate for the hypothesis being tested?

-Is the sample size sufficient to ensure adequate power to address the hypothesis being tested?

-Were correct statistical analysis used to support conclusions?

-Are there concerns about ethical or regulatory requirements being met?

Reviewer #2: (No Response)

Reviewer #3: (No Response)

Reviewer #5: -

**Results**

-Does the analysis presented match the analysis plan?

-Are the results clearly and completely presented?

-Are the figures (Tables, Images) of sufficient quality for clarity?

Reviewer #2: (No Response)

Reviewer #3: (No Response)

Reviewer #5: -

**Conclusions**

-Are the conclusions supported by the data presented?

-Are the limitations of analysis clearly described?

-Do the authors discuss how these data can be helpful to advance our understanding of the topic under study?

-Is public health relevance addressed?

Reviewer #2: (No Response)

Reviewer #3: (No Response)

Reviewer #5: -

**Editorial and Data Presentation Modifications?**

Reviewer #2: (No Response)

Reviewer #3: (No Response)

Reviewer #5: -

**Summary and General Comments**

Reviewer #2: (No Response)

Reviewer #3: This revised version has answered the questions I raised very well， I have no further concerns.

Reviewer #5: The authors are still confused by the word "mimic", which means "to closely resemble".

The phrase in the abstract, "exhibited a better correlation with leptospires grown in WB" is appropriate.

PLOS authors have the option to publish the peer review history of their article (what does this mean? ). If published, this will include your full peer review and any attached files.

**Do you want your identity to be public for this peer review?** For information about this choice, including consent withdrawal, please see our Privacy Policy .

Reviewer #2: No

Reviewer #3: **Yes:**  Zhe Wang

Reviewer #5: No

---

## [Editor Report · Acceptance letter]

Dear Dr. Wunder Jr,

We are delighted to inform you that your manuscript, "DMEM and EMEM as alternate growth media for pathogenic Leptospira," has been formally accepted for publication in PLOS Neglected Tropical Diseases.

Best regards,

Shaden Kamhawi

co-Editor-in-Chief

Paul Brindley

co-Editor-in-Chief
